# Cardiovascular Disorders Triggered by Obstructive Sleep Apnea—A Focus on Endothelium and Blood Components

**DOI:** 10.3390/ijms22105139

**Published:** 2021-05-12

**Authors:** Jakub Mochol, Jakub Gawrys, Damian Gajecki, Ewa Szahidewicz-Krupska, Helena Martynowicz, Adrian Doroszko

**Affiliations:** Department of Internal Medicine, Hypertension and Clinical Oncology, Faculty of Medicine, Wroclaw Medical University, Borowska 213, 50-556 Wroclaw, Poland; jakub.mochol@student.umed.wroc.pl (J.M.); jakub.gawrys@umed.wroc.pl (J.G.); damian.gajecki@umed.wroc.pl (D.G.); ewa.szahidewicz-krupska@umed.wroc.pl (E.S.-K.); helena.martynowicz@umed.wroc.pl (H.M.)

**Keywords:** obstructive sleep apnea (OSA), endothelial dysfunction (ED), oxidative stress, nitric oxide (NO), asymmetric dimethylarginine (ADMA)

## Abstract

Obstructive sleep apnea (OSA) is known to be an independent cardiovascular risk factor. Among arousal from sleep, increased thoracic pressure and enhanced sympathetic activation, intermittent hypoxia is now considered as one of the most important pathophysiological mechanisms contributing to the development of endothelial dysfunction. Nevertheless, not much is known about blood components, which justifies the current review. This review focuses on molecular mechanisms triggered by sleep apnea. The recurrent periods of hypoxemia followed by reoxygenation promote reactive oxygen species (ROS) overproduction and increase inflammatory response. In this review paper we also intend to summarize the effect of treatment with continuous positive airway pressure (CPAP) on changes in the profile of the endothelial function and its subsequent potential clinical advantage in lowering cardiovascular risk in other comorbidities such as diabetes, atherosclerosis, hypertension, atrial fibrillation. Moreover, this paper is aimed at explaining how the presence of OSA may affect platelet function and exert effects on rheological activity of erythrocytes, which could also be the key to explaining an increased risk of stroke.

## 1. Introduction

Obstructive sleep apnea (OSA) is characterized by recurrent obstruction of the upper airway during sleep causing intermittent hypoxia (IH). The prevalence is increased by advanced age, male sex, higher body mass index and ranges, according to some estimations, from 9% to 38% in general population [1]. OSA is known to be an independent cardiovascular risk factor. The incidence during three years observation of cardiovascular mortality, myocardial infarction, stroke, and unplanned revascularization in patients undergoing percutaneous coronary intervention was higher in the OSA group (18.9% versus 14.0% in the non-OSA group) [2]. Numerous possible mechanisms contributing to the progression of cardiovascular disorders remain in the focus of interest (Figure 1).

Among increased thoracic pressure and activation of the sympathetic system, intermittent hypoxia (IH) is now being recognized as a potential major factor contributing to the pathogenesis of OSA-related comorbidities. IH is characterized by cycles of hypoxemia followed by reoxygenation that contribute to the development of the ischemia-reperfusion injury [3]. The biochemical consequences of hypoxia comprise the inhibition of the Krebs cycle and promotion of lactate synthesis. The impairment of mitochondrial oxidative phosphorylation in the course of hypoxia, followed by subsequent reoxygenation, induces the production of reactive oxygen species (ROS). ROS generation involves the mitochondrial respiratory chain and numerous enzyme complexes, including the NADPH oxidase, nitric oxide (NO) synthase and the xanthine oxidase [4].

Oxidative stress results from an imbalance between the pro-oxidants formation and neutralization. Reactive oxygen species (ROS) are toxic highly reactive and unstable compounds. Superoxide reacts with NO creating peroxynitrite (ONOO^−^) which is a very potent entity, being ~1000× stronger as an oxidizing agent than is H_2_O_2_. The ONOO^−^ can influence posttranslational protein modifications altering their structure, activity and function, leading in turn to impairment of signaling pathways. Markers of ONOO^−^ formation (such as nitrotyrosines or isoprostanes) can be found in many disease states including brain injury [5], heart injury [6], preeclampsia [7], and inflammation [8].

Reactive oxygen species (ROS) can limit NO bioavailability by reacting with cofactors of NO synthase (NOS). ROS cause depletion of tetrahydrobiopterin and alter the ratio of oxidized to reduced glutathione inducing NOS S-glutathionylation [9]. Oxidative stress is strongly attributed to endothelial NOS dysfunction (eNOS uncoupling). Converted to a superoxide-producing enzyme, uncoupled eNOS not only leads to reduction of the NO generation but also potentiates the preexisting oxidative stress. An understanding of the biology of NO, O2^−^ and ONOO^−^ as well as the importance of the NOS uncoupling in physiological and pathological setting are crucial for understanding the nitrosative stress-related controversies. There is a critical balance between cellular concentrations of NO, O2^−^, and superoxide dismutase, which physiologically favors NO production but in pathological conditions such as ischemia/reperfusion(I/R) results in ONOO^−^ generation.

The NO bioavailability can be also diminished in other possible mechanism involving asymmetric dimethylarginine (ADMA) and monomethylated arginine (L-NMMA) which are endogenous competitive inhibitors of the NOS. ADMA plays a role in the development of endothelial dysfunction and is considered as a marker of oxidative stress. Most importantly, OSA is associated with a decrease in the NO bioavailability and changes in the platelet function and erythrocyte rheological disturbances [10].

## 2. Molecular Consequences of Hypoxia/Reoxygenation (H/R) on Endothelial Function in OSA before CPAP Treatment 

There are many mechanisms triggered by the hypoxia/reoxygenation injury. Most of them are associated with ROS overproduction and cellular damage. The impact of hypoxemia causing endothelial dysfunction was examined in numerous animal and human in vitro and in vivo models. Intermittent hypoxia in rats impaired endothelial function by attenuating the integrity of endothelium and lowering the number of endothelial progenitor cells (EPCs) in the blood [11]. Endothelial progenitor cells (EPCs) are circulating bone marrow-derived precursors which are capable of excreting microvesicles (MVs) containing gene messages (mRNAs and miRNAs). MicroRNAs (miRNAs) are small non-coding RNAs which play a role in several cellular processes. Not only progenitor cells, but also activated endothelial cells and white blood cells are capable of producing the microparticles. MVs may have either beneficial or detrimental effects on endothelial cells. In the medium containing tumor necrosis factor α (TNFα) they activate the caspase 3 which leads to apoptosis. MVs transduce information associated with ROS production, inducing angiogenesis and activation of the PI3K/eNOS/NO pathway [12]. MVs released during hypoxia/reoxygenation injury are pro-apoptotic and pro-oxidative [13]. Another study confirms that circulating MVs cause increased permeability and disruption of tight junctions along with increased adhesion molecule expression, reduce eNOS expression and promote increased monocyte adherence. Comparing the influence of microvesicles isolated before and after PAP therapy, the disturbances in endothelial cells function such as increased permeability and disruption of tight junctions were attenuated with treatment [14].

Priou et al. found higher levels of microvesicles derived from granulocytes and activated leukocytes (CD62L+) in patients with the oxyhemoglobin desaturation index (ODI) ≥ 10. MVs have increased expression of endothelial adhesion molecules (E-selectin, ICAM-1, Integrin alpha-5) and cyclooxygenase 2.

MVs from desaturating patients injected into mice impaired the endothelium-dependent relaxation of vascular smooth muscle cells (VSMCs) in aorta and the flow-mediated dilation (FMD) in small mesenteric arteries resulting from decreased NO production. The endothelial NO synthesis negatively correlated with the number of active leukocytes and the sleep apnea severity. In vitro, MVs from desaturating patients reduced endothelial NO production by enhancing phosphorylation of eNOS at the site of inhibition and increasing expression of caveolin-1. Caveolin-1 is a membrane protein which regulates endothelial nitric oxide synthase (eNOS) activity and takes part in cellular insulin-signaling. In the study by Sharma et al. chronic 3-day intermittent hypoxia (IH) exposure on human coronary artery endothelial cells increased caveolin-1 and endothelin-1 expression resulting in decreased NO bioavailability [15,16].

Skin biopsies obtained from OSA patients with severe nocturnal hypoxemia demonstrate a significant upregulation of eNOS, TNFα-induced protein 3, hypoxia-inducible factor 1 alpha (HIF-1α), vascular endothelial growth factor (VEGF) and vascular cell adhesion molecule 1 (VCAM-1) [17]. The expression of endothelial-cell-specific molecule-1 (ESM-1, Endocan), VEGF and HIF-1α was also significantly increased in the human umbilical vein endothelial cells (HUVEC) subjected to IH and in patients with OSA. ESM-1 is upregulated by the HIF-1α/VEGF pathway under IH in endothelial cells, playing a critical role in enhancing adhesion between monocytes and endothelial cells [18]. IH increased advanced glycation end products formation and activated NF-кB signaling in monocytes, resulting in enhanced monocyte adhesion, chemotaxis, and promoted macrophage polarization toward a pro-inflammatory phenotype [19]. Enhancing adhesion and infiltration activity monocyte chemoattractant protein-1 (MCP-1) was also increased in monocytes under IH [20]. Moreover increased macrophage population with pro-inflammatory expression of CD36 and Ly6c was confirmed in the aortic wall in a murine model of IH [21]. In mice nocturnal intermittent hypoxia increased mRNA levels of 5-lipoxygenase and CysLT1 receptor, which was strongly associated with atherosclerosis lesion size. That confirms cysteinyl-leukotrienes (CysLT) pathway activation as a potential mechanism responsible for developing atherosclerosis connected with IH [22]. Another study trying to explain OSA-dependent atherogenesis found an increased expression of toll-like receptors (TLRs) and receptor for advanced glycation end-products (RAGE) in atherosclerotic plaques from patients with severe OSA [23].

An in vitro model of OSA shows that endothelial cells originating from distinct vascular beds respond differently to intermittent hypoxia. In human dermal microvascular endothelial cells IH decreased the expression of eNOS and HIF-1α, while in coronary artery endothelial cells HIF-1α expression was increased [17]. The HIF-1α activates the NFκB, a transcription factor which regulates several pro-inflammatory genes, including TNFα, interleukin (IL)-8, and IL-6 [24]. IL-6, the epidermal growth factor family ligands, and tyrosine kinase receptors induced by IH may be involved in the proliferation of vascular smooth muscle cells [25]. Another study showed that a further vascular and cardiac dysfunction in mice under IH can be triggered by trombospondin-1 though cardiac fibroblast activation and increasing angiotensin II activity [26].

In conclusion, OSA may result in endothelial dysfunction by limiting the NO availability, promoting oxidative stress, up-regulating the expression of pro-inflammatory cytokines. Microparticles and signaling factors lead to lower permeability of endothelial cells, enhanced adhesion and increased apoptosis (Figure 2).

## 3. Endothelial Function after Treatment with CPAP 

An appropriate CPAP therapy in randomized control trials conducted on OSA patients improved flow mediated dilation (FMD), suggesting its potentially beneficial role in cardiovascular risk reduction [27,28]. A meta-analysis confirms that CPAP increases the absolute FMD value by a mean of 3.87% [29]. Flow mediated dilation (FMD) measures the change of the brachial artery diameter after a brief period of forearm ischemia. It helps to assess endothelial function and its ability to dilate the vessel by producing NO and prostacyclin. There are many other physical methods to assess endothelial function including venous occlusion plethysmography, peripheral arterial tonometry (PAT) and optical techniques using laser doppler flowmetry (LDF) that can be coupled with provocation tests (post-ischemic hyperemia, local heating). However, more studies are needed in order to validate their usefulness in assessing endothelial dysfunction.

As far as the literature is concerned, the effect of CPAP on inflammatory reaction is not proven yet. Inflammatory markers such as IL-8, hs-CRP, and TNF-α did not significantly change from baseline after 1 year CPAP therapy [30]. However, the use of CPAP for at least 5 h per night decreased TNF-α levels in women suffering from OSA [31]. Interestingly, CPAP tends to lower TNF-α, which is initially increased in OSA patients [32]. These diverse effects of CPAP treatment could be explained by a low adherence to the therapy and the fact that better effects are observed with the therapy duration of ≥3 months and more adequate compliance (≥4 h/night).

Another positive effect after 12 weeks of CPAP therapy was a decreased expression of angiotensin receptors type-1 (AT-1R) measured in the gluteal subcutaneous tissue [33]. The AT-1R mediates the major cardiovascular effects of angiotensin II, including cardiac hypertrophy, augmentation of peripheral noradrenergic activity and vascular smooth muscle cells proliferation.

Three months of CPAP treatment significantly increased the level of sirtuin 1 (SIRT1) and serum levels of NO derivative in the blood [34]. SIRT1 is a histone/protein deacetylase which regulates the eNOS, restores the NO availability and is involved in different aspects of aging, metabolism, stress resistance and cardiovascular disease. 

Patients with OSA showed a higher adventitial vasa vasorum density, correlating with AHI [35]. This could be explained by increased VEGF activity induced by IH. One meta-analysis confirmed that CPAP therapy improved endothelial function associated with VEGF lowering [36]. On the contrary, short return of OSA using sham CPAP for 2 weeks was not associated with changes in endocan, ET-1, resistin and VEGF. However, a significant decrease in vasodilatory peptide adrenomedullin was found [37,38]. Adrenomedullin is a protective endothelial product stimulated by IH, which could partially explain why the CPAP therapy may deteriorate endothelial function or exert a neutral effect in the short-term observational studies.

In the moderate to severe OSA, the 2-month CPAP treatment vs. sham did not reduce the plasma concentrations oxidative stress-related markers [39]. In a study by Borges comparing the 8-week CPAP therapy with aerobic training, no significant changes regarding oxidative stress markers and cell-free DNA levels were detected [40]. Interesting outcomes were shown in a mice model study, where the animals treated with a high-fat diet revealed a positive effect of the low-frequency hypoxia. The serum levels of the oxidative stress markers were increased in the mice treated with a high-frequency intermittent hypoxia (60 hypoxic events/h) and decreased by treating with a low frequency hypoxia (10 events/h) [41]. That could be partially explained by the activation of protective mechanisms during IH.

Although there is substantial evidence that CPAP improves endothelial function (Figure 3), the antioxidant capacity is not changed significantly. The median plasma nitrite level and total antioxidant status did not show any significant difference between the OSA and the control groups. Nevertheless, the oxidant-antioxidant balance was shifted toward the oxidant side in OSA cases [42].

## 4. Endothelial Function after Treatment with CPAP in Specific Subgroups 

The molecular background of the tissue damage in the course of hypoxia and the hypoxia followed by reoxygenation has already been studied at molecular and functional level in numerous basic science-based studies. Since the presence of obstructive sleep apnea might be easily mimicked by the episodes of recurrent tissue hypoxia in the long-term observation, its treatment with the CPAP, leading to the improvement of the oxygen supply does not represent in fact easily a reoxygenation injury. Once the CPAP treatment begins, the hypoxia episodes do not appear or are much less profound, which is accompanied by an optimal oxygen supply preventing from reoxygenation injury. Hence, the clinically observed OSA followed the onset of its treatment mimics hypoxia-induced injury accompanied by initial reoxygenation injury, but its treatment with CPAP reflects rather its prevention and “wound healing” in the long-term outcome. Therefore, the use of CPAP cannot be simply attributed to the reversal of all molecular changes observed in the studies on H-R injury, and its effects may not inhibit or reverse simply all the changes observed in the chronically reversible H-R conditions. The following sections comment on the clinical studies on subjects with the OSA treatment aiming at the explanation of the molecular background of the putative therapeutic effect of CPAP on the overall cardiovascular risk (Table 1). 

### 4.1. Atherosclerosis

Patients with the apnea-hypopnea index ≥20 showed an increased risk for arterial stiffness correlated with the arousal index and with mean O2 saturation compared to other poststroke patients with similar age, sex, body mass index, hypertension and diabetes mellitus status. However, no significant differences were seen in endothelial function measured by Endo-PAT 2000 in patients suffering from sleep apnea [43]. Another study including elderly subjects did not detect significant differences of pulse wave velocity depending OSA severity [44].

In a metanalysis, the pulse wave velocity was more associated with age, systolic blood pressure and diabetes than with the apnea parameters [45].

The coronary artery plaque burden was significantly associated with AHI and was independent of other traditional cardiovascular risk factors. AHI index correlates better with more advanced coronary atherosclerosis than the severity of arterial desaturation [46]. In another study moderate/severe OSA was associated with 10% lower hyperemia index measured using the Endo-PAT device and 35% higher coronary artery calcium (CAC) quantified by electron beam computed tomography, which did not reach the statistical significance (*p* = 0.08 for both comparisons) [47]. Participants with moderate-to-severe OSA were 1.6x more prone to have an ascending thoracic aorta calcification than those without OSA, and the calcification was greater in patients with higher epicardial fat volume [48]. Additionally, the non-dipper profile of nocturnal hypertension makes the OSA patients more prone to a high-risk atherosclerosis [49].

There were different studies results regarding carotid intima media complex. Carotid IMT was not increased in adults with moderate to severe OSA versus controls and does not change following 4 months of PAP treatment in one study [50]. On the contrary, in the study of Catala, the carotid IMT decreased markedly in the CPAP group [51]. In a metanalysis CPAP had no impact on carotid IMT in OSA patients, carotid IMT was significantly decreased after CPAP treatment in more severe OSA patients and patients with long CPAP usage [52].

### 4.2. Myocardial Infarction

Hypoxia-reoxygenation (H/R) injury is observed during an early phase of myocardial infarction and at the beginning of reperfusion therapy as well as in the severe OSA desaturations, which mimics asphyxia (hypoxia). The increased ONOO^-^ formation in the heart during H/R, may change several proteins which compose the contractile machinery of the heart leading to systolic dysfunction and cardiac injury [53]. Peroxynitrites formed during the injury may lead to the nitration of cardiac contractile proteins (including the myosin light chain 1 and 2, MLC1 and MLC2) leading to their increased susceptibility to subsequent proteolysis by the matrix metalloproteinases (i.e., MMP-2) [54,55] Similar changes were observed in an ex vivo model of myocardial infarction [56], pointing thus at the same cardiotoxic pathophysiological mechanisms of the ischemia/reperfusion and hypoxia/reoxygenation cardiac injury. In that studies, compensatory increase in the tissue inhibitor of matrix metalloproteinases-4 (TIMP-4) expression was observed during ischemia, but not reperfusion, which reflects its role in the ischemic preconditioning.

In clinical setting, the prevalence of moderate/severe OSA in patients with coronary artery disease was associated with diminished plasma level of C1q/TNF-related protein-9 (CTRP9) [57]. CTRP9 is an adipokine that protects the heart against ischemic injury and ameliorates cardiac remodeling, which in turn could explain the role of OSA in exacerbating the coronary artery disease.

The role of OSA in increased incidence of myocardial infarction can be also considered together with the progression of atherosclerosis and altered platelet function. The role of CPAP effectiveness in reducing the myocardial infarction incidence is not proven yet. The ISAACC study showed that among non-sleepy patients with acute coronary syndrome, the presence of OSA was not associated with an increased prevalence of cardiovascular events and treatment with CPAP did not significantly reduce this prevalence [58]. On the contrary, RICCADSA trial confirmed that CPAP treatment may reduce this risk, if the device is used at least 4 h/day [59].

### 4.3. Heart Function

Advanced and untreated sleep apnea, where the recurring prolonged periods of hypoxia followed by reoxygenation mimic to some extend the pathophysiological cascade observed in the course of asphyxia may deteriorate heart function. 

Among men 40 to 70 years old, those with AHI > or = 30 were 68% more prone to develop coronary artery disease than those with AHI < 5 [60]. Bakker et al. study showed that the 3 months of CPAP therapy resulted in lowering the left ventricle (LV) end-diastolic volume, as assessed by the magnetic resonance imaging (MRI) [61]. Conversely, in another study no significant changes were noted in ventricular dimensions, systolic and diastolic function, valvular function and coronary vasodilation to nitroglycerin after 3 months of CPAP. The OSA patients display right ventricle dilatation and an increased wall thickening (eccentric hypertrophy) [62].

The coronary flow reserve (CFR), which is decreased in patients with moderate to severe OSA, improved after 3 months of CPAP [63]. CPAP treatment in subjects with congestive heart failure may achieve symptomatic and functional improvements, but exercise alone improved quality of life more than CPAP.

Interestingly, another type of positive pressure therapy (adaptive servo-ventilation therapy) is contraindicated in patients with ejection fraction ≤ 45%. Although the SERVE-HF control trial showed increased mortality of any cause, there is no clear mechanism of adverse effects of the therapy. There is a hypothesis that an increased ventilation could cause alkalosis which may interfere with the ion transmembrane transport resulting in proarrhythmic action.

### 4.4. Diabetes

The common synergism for glucose intolerance, insulin resistance, developing diabetes and OSA can multiply the negative effect of endothelial dysfunction and may increase a cardiovascular risk significantly. Oxidative stress, protein glycation, impairment of the NO bioavailability can be a link multiplying endothelial dysfunction in comorbidity of OSA and diabetes. Molecular changes are partially reversible by the CPAP (Continuous Positive Airway Pressure) therapy, lowering the BMI and lifestyle changes.

Diabetes can exacerbate endothelial dysfunction through ADMA generation stimulated by glyceraldehyde-derived advanced glycation end products (glycer-AGEs). Dyslipidemia works through oxidized LDL, which stimulates endothelial cell inflammation, oxidative stress, and apoptosis.

Lifestyle intervention and successful weight reduction significantly improved AHI, BMI, serum triglycerides and insulin resistance in mild OSA patients [64]. The CPAP effect on improving endothelial function measured by the FMD is even greater in OSA patients with coexisting diabetes [61]. The treatment efficiency in coexisting diabetes and OSA could be explained by common mechanisms leading to endothelial dysfunction, which comprises ROS overproduction. In diabetes, ROS are generated as an effect of increased expression of NADPH oxidase, cyclooxygenase, lipoxygenase and impaired antioxidant defense. Similar to OSA, the pathophysiological role of microparticles and increased inflammatory markers are observed. The benefit of CPAP therapy on endothelial function may be explained by similar mechanism in both diseases.

The treatment with CPAP after 12 or 24 weeks showed no effectiveness in changing glycated hemoglobin (HbA1_c_) levels [65]. CPAP treatment significantly improved the HOMA index, but no significant changes in fasting glucose were observed [66]. CPAP has a favorable effect on insulin resistance, but it is not associated with any significant changes in the total adiponectin levels [67]. Prospective studies showed that regular CPAP use was associated with reduction of diabetes incidence from 3.41 to 1.61 per 100 person-years [68].

### 4.5. Hypertension

Sympathetic hyper-activation and alteration in the renin-angiotensin-aldosterone axis may play a pathophysiological role in patients with obstructive sleep apnea. Increased renin generation is induced by efferent renal sympathetic nerve activation and leads to activation of the renin-angiotensin-aldosterone system (RAAS). OSA causes systemic inflammation and oxidative stress, which results in increased endothelin-1 generation and decreased NO production in endothelial cells. 

Endothelin receptor antagonist, bosentan, suppressed the increase in SBP during a 5 min hypoxic challenge (143 ± 5 mmHg vs. 127 ± 3 mmHg), which confirms the role of endothelin in response to acute hypoxia in patients with severely untreated OSA [69].

OSA is related to an increased risk of resistant hypertension [70]. In patients with resistant hypertension and OSA catheter-based renal sympathetic denervation after 6 months reduced the mean ambulatory blood pressure by 8.3/6.2 mmHg, with no significant changes in the sleep apnea severity [71].

CPAP treatment may improve the hypertension and cardiovascular outcomes by reducing aldosterone excess in resistant hypertensive individuals with OSA. There was a borderline significant reduction in 24 h urine collection for aldosterone after 6 months of follow-up [72].

Although it is well known that the treatment of sleep apnea lowers blood pressure, the effectiveness of the therapy is not very high, but its effect translated into CV events and mortality reduction over the long term, is not completely negligible. After 6 months, the CPAP caused greater reduction of night-time systolic blood pressure at 4.7 mm Hg. The CPAP was associated with significant reductions of the 24 h ambulatory systolic blood pressure of 2.32 mm Hg and diastolic blood pressure of 1.98 mm [73,74]. A better effect of CPAP was observed in the resistant hypertension subjects −5.40 mmHg and −3.86 mmHg respectively [75]. In the subjects with severe oxygen desaturations (SpO2 < 77%) with good CPAP adherence the reduction of systolic blood pressure was observed [76]. Severe desaturations but not AHI were associated with better hypotensive response of CPAP. That could be explained by reducing the ROS production during hypoxia/reoxygenation injury. Indication for the therapy should comprise not only not only the frequency but also the depth and duration of sleep-related upper airway obstructions. Another aspect is that deeper hypoxia leads to stronger stimulation of the peripheral chemoreceptors, which increase sympathetic system. Other study showed that a 2 week withdrawal of CPAP leads to a relevant increase in morning blood pressure of nearly 10 mm Hg in the moderate to severe sleep apnea patients [77]. Interestingly supplemental oxygen, which reduced intermittent hypoxia (IH) and had a minimal effect on AHI abolished the rise in morning blood pressure during CPAP withdrawal [78]. Therefore, IH, and not recurrent arousals, appears to be the dominant cause of daytime increases in blood pressure in OSA. There is a need to better understand the effects of CPAP on BP since a better characteristic of patients who benefit most might help to tailor therapy according to the expected benefit to reduce BP and in general improve the patient’s cardiovascular risk profile. Interestingly, a systematic review and meta-analysis of randomized controlled trials (RCTs) compared the effect of CPAP on BP in particular subgroups of patients and aimed at defining the group with the best response to treatment. The meta-analysis has defined younger age, uncontrolled blood pressure and severe OSA-related oxygen desaturations as positive predictors of a favorable blood pressure response to OSA treatment. Desaturation of less than 77% were associated with a greater BP drop at follow-up in treated patients further supporting the role of intermittent hypoxia in the pathogenesis of OSA- related hypertension [76]. This observation points at oxidative stress as on the important therapeutic target.

### 4.6. Pulmonary Hypertension

The prevalence of pulmonary hypertension in OSA ranges from 17 to 53% [79]. Small pulmonary arteries constrict in the presence of alveolar hypoxia. It helps to redirect blood flow from poorly-ventilated lung regions to those which are well-ventilated. It is also known as hypoxic pulmonary vasoconstriction (HPV) or the van Euler–Liljestrand mechanism. During global alveolar hypoxia, HPV leads to pulmonary hypertension. ROS originating during hypoxia/reoxygenation in OSA patients can interact with protein kinases, phospholipases, and other ion channels modulating response of the pulmonary arterial smooth muscle cells [80]. The patients suffering from OSA had increased serum levels of C-reactive protein and 8-isoprostane, TNFα, interleukin (IL)-1β and IL-6 in the pulmonary tissue [81]. Another study revealed that limited NO bioavailability caused by IH could be compensated by increased pulmonary vascular smooth muscles sensitivity to NO and cGMP [82].

In one study the CPAP therapy was associated with a significant decrease in pulmonary artery pressure in patients with isolated OSA and pulmonary hypertension [83].

### 4.7. Atrial Fibrillation and Other Arrhythmias

Chronic OSA induces sympathetic activation followed by a structural and electrical remodeling of the atria contributing to the AF maintenance and recurrence of AF paroxysms [84]. Higher AHI is a known factor associated with persistent or permanent AF [85].

Reactive oxygen species and chronic inflammation decrease Na^+/^K^+^ ATPase currents, alter the Ca2^+^ homeostasis and down-regulate some proteins, such as connexin-43 [86]. Changed ion currents contribute to increased proarrhythmic activity. 

The effects of a 3-month CPAP therapy were observed in patients with arrhythmias such as supraventricular and ventricular extrasystoles, atrial fibrillation, non-sustained ventricular tachycardia (nsVT), and sinus pauses [87]. There was a significant decrease in atrial and ventricular ectopy count in patients with AF [88]. Another study confirms that CPAP therapy has an impact on reversing the atrial remodeling in patients with OSA. Following the treatment, the atrial pressure, volume overload and serum BNP levels were significantly reduced. These observations may suggest that the substrate predisposing to AF may be reversible and measured by total atrial conduction time assessed by tissue doppler imaging (PA-TDI interval) and BNP [89].

CPAP therapy resulted in a higher AF-free survival rate and an AF-free survival off antiarrhythmic drugs or repeat ablation. AF recurrence rate of CPAP-treated patients was similar to a group of patients without OSA and was significantly higher in CPAP nonuser patients [90]. Patients with OSA are less likely to remain in sinus rhythm after catheter ablation of AF. Concomitant OSA increased (HR 2.61) and usage of CPAP therapy decreased (HR 0.41) the probability of AF recurrences in prospective study [91]. On the contrary, randomized controlled trial assessing the impact of treatment of OSA on recurrence of AF after direct current cardioversion (DCCV) did not detect a difference between those treated with PAP versus usual care [92].

Another study found that apnea/hypopnea duration is the main factor for heart conduction disorders [93]. Cardiac activity pauses were correlated with the longest apnea, as well as the AHI and oxygen desaturation index [87].

### 4.8. Pediatric Population—Effect of Tonsillectomy

Interesting studies showed that OSA in children can induce endothelial dysfunction. The lowest FMD values were found in children with higher AHI [94]. The improvement in FMD following adenotonsillectomy was found together with a decrease in oxidative stress [95]. Hypermethylation of the core promoter region of eNOS gene in the OSA children was related to decreased eNOS expression. Additionally, children with OSA had increased expression of genes encoding pro-oxidant enzymes and decreased expression of genes encoding anti-oxidant enzymes [96]. OSA in adolescents appears to increase independently the risk of dyslipidemia, insulin resistance, and hypertension [97].

**Table 1 ijms-22-05139-t001:** Endothelial function after treatment in specific subgroups.

OSA Subpopulation	Demonstrated Molecular Pathomechanism	Effect of OSA Treatment
Atherosclerosis	endothelial dysfunction	No effect on endothelium, did not reduced PWV [44]
		(2019, clinical trial, 101 patients)
Myocardial Infarction	Increased peroxynitrite formation [53], nitration of cardiac contractile proteins (MLC1 and MLC2) and their subsequent degradation (by MMP-2) protective role of TIMP-4 in ischemic preconditioning [54,55,56]	The ISAACC—among non-sleepy patients with acute coronary syndrome, treatment with CPAP did not significantly reduce the prevalence of acute coronary syndromes [58].
		(2020, randomized controlled trial, 1264 subjects)
		On the contrary, RICCADSA trial confirmed that CPAP treatment may reduce this risk, if the device is used at least 4 h/day [59]
		(2016, randomized controlled trial, 244 subjects)
Heart Failure	Increased peroxynitrite formation [53]	No effect on endothelium, lowering the left ventricle end-diastolic volume [61]
		(2020, randomized controlled trial, 141 patients)
Diabetes	impairment of the NO bioavailability, ROS	improved HOMA index, no effect on adipokine level [67]
		(2015, meta-analysis)
Hypertension	activation of RAAS	SBP—2.32 mm Hg [74]
		(2015, meta-analysis, 794 patients)
Pulmonary Hypertension	Increased inflammatory cytokines [81]	decrease in pulmonary artery pressure [83]
		(2010, metanalysis, 222patients)
Atrial Fibrillation	down-regulation connexin-43 [86]	(HR 0.41) the probability of AF recurrences [91]
		(2012, prospective study, 153 patients)
Children	decreased eNOS expression [96]	FMD improvement after tonsillectomy [95]
		(2015, clinical trial, 144 patients)

## 5. Platelet Function

OSA may change the platelet function and contribute to the pro-coagulative state. Platelets play a great role in atherothrombosis and their reactivity appeared to be higher in OSA patients (Table 2). 

There were several studies observing that mean platelet volume (MPV) was independently correlated with AHI. MPV was considered as a new marker associated with atherothrombosis. An increased MPV previously was explained by new generated platelets, which have higher volume and density. The new data suggest that there are different platelets subpopulations originating from different megakaryocytes. The high-volume platelets correlate with higher expression of adhesion molecules, increased aggregation, enhanced release of thromboxane TXA2, whereas small and low-density platelets have an enhanced intracellular Ca2^+^ response to thrombin. MPV and platelet distribution width (PDW) are higher in severe OSA compared to control group, but no significant differences between controls and patients with mild and moderate OSA were observed. Mean platelet volume and red blood cell distribution width changed significantly after 3-month CPAP treatment [98,99].

Arachidonic acid- and adenosine diphosphate (ADP)-induced aggregation in the OSA group was significantly higher than in the non-OSA group [100]. In experimental human and animal models induced hypoxia/reoxygenation contributed to enhanced TXA2 formation and increased activation of matrix metalloproteinase 2 (MMP-2) leading to platelet activation and subsequent aggregation [101,102].

Moreover, patients with OSA were characterized by significantly lower inhibitory rate of the ADP-dependent aggregation. OSA patients were more likely to have high residual platelet reactivity after acetylsalicylic acid or clopidogrel therapy [98]. OSA-related intermittent hypoxia and reoxygenation frequency contributed to platelet hyperaggregability for ADP more than total hypoxic time during the sleep. Patients with one or more vascular risk factors such as diabetes, hypertension, smoking, hyperlipidemia were prone to platelet hyperaggregability for both ADP and collagen. After three months CPAP treatment partially normalized the OSA-related ADP- and collagen-induced platelet hyperaggregability [103]. The possible mechanisms underlying this phenomenon include normalized levels of proinflammatory mediators and decreased activity of sympathetic nervous system after treatment.

Another study showed that desaturating patients had lower GPIb fluorescence in circulating platelets. The platelet surface P-selectin, platelet surface-activated GPIIb/IIIa, platelet-monocyte aggregation, platelet-neutrophil aggregation, CD62P and were not significantly correlated with markers of OSA in one study [104]. Conversely, another study found a progressive increase in the concentrations of soluble E-selectin, P-selectin and L-selectin in OSA patients [105]. Even though the surface selectins remain unchanged, the increased soluble forms may be considered as markers of inflammation of vascular wall with prothrombic activity. Another study confirms that serum content of platelet P-selectin and P-selectin glycoprotein ligand 1 are increased proportionally with OSA severity [106]. CPAP significantly decreased serum levels of sCD40L and sP-selectin in patients with moderate to severe OSA [107,108].

### Hypercoagulative State in OSA

Patients with moderate to severe OSA have elevated PT and INR compared with healthy individuals [99]. This could be connected with increased activity of coagulation factor VII.

OSA patients are characterized by elevated plasma fibrinogen levels which is induced by chronic inflammation, exaggerated platelet activity, and reduced fibrinolytic capacity. In hypertensive patients with OSA fibrin clot was characterized by more compact fibrin structure, impaired fibrinolysis and faster clot formation, which was normalized after 3 months of CPAP treatment [109].

**Table 2 ijms-22-05139-t002:** Cardiovascular thromboembolic disorders and CPAP effectiveness.

Demonstrated Platelet-Derived Molecular Pathomechanism	Thromboembolic Disorders	Clinical Effect of OSA Treatment-Based on Clinical Studies
-platelet hyperaggregability for both ADP and collagen	Stroke	Reduction in risk of stroke in elderly patients [110] (2021, retrospective cohort study, 5757 patients)
-lower inhibitory rate of the ADP-dependent aggregation	Myocardial Infarction	No significant effect in the MI incidence reduction in prospective observation of the CPAP treatment patients [58] (2020, randomized controlled trial, 2551 patients), [111] (2016, randomized controlled trial, 2717 patients)
-high residual platelet reactivity after acetylsalicylic acid or clopidogrel therapy [98]		

## 6. Erythrocytes

The cause of an increased cardiovascular morbidity in OSA patients could be found in functional and structural changes of erythrocytes. OSA is associated with an increased erythrocyte adhesion measured by the optic protocols on glass slide samples. Erythrocyte adhesiveness and aggregation correlate with an increase of inflammatory acute phase proteins [112].

Asymmetric dimethylarginine (ADMA) and symmetric dimethylarginine (SDMA) are endogenous inhibitors of NO synthesis whose plasma concentrations were demonstrated to be elevated in OSA patients. Plasma ADMA levels were associated with a lower membrane fluidity of erythrocytes, suggesting that ADMA might have a close correlation with the rheologic behavior of erythrocytes and the microcirculation [113]. ADMA and the non-specific pharmacological NOS inhibitor, L-NAME, independently reduced the deformability of red blood cells (RBC) obtained from rabbits treated with a high cholesterol diet. The effect was reversed with activators of the NO pathway. These results suggest that NO plays an important role in improving the microcirculation by restoring RBC deformability. Impaired erythrocyte deformability may be partially dependent upon the accumulation of ADMA in RBC [114]. Red blood cells contain a nitric oxide synthase (RBC-NOS) which produces NO modifying RBC deformability through direct S-nitrosylation of cytoskeleton proteins including spectrin-α, spectrin-β [115]. Other cytosolic erythrocytic NG-dimethylated proteins like protein 4.1 could also play a role in changing the stability of the erythrocyte membrane.

NO binds cooperatively to βCys93 in oxygenated Hb creating s-nitrosyloheamoglobin (SNO-Hb). In the deoxygenated state of hemoglobin reactions of NO with heme-iron are favored over thiol. Only a small fraction of the NO carried by Hb is released from RBCs, but transition from high to low oxygen tension in the peripheral arterioles and capillaries promotes its release as SNO-based vasodilatory activity [116]. No data was found on the role of OSA and ROS in the hemoglobin S-nitrosylation.

Moreover, RBCs contain the arginine-rich proteins which could be a potential source of inhibitory methylarginines (Figure 4). Protein-arginine methyltransferases (PRMTs) catalyze the methylation of proteinic L-arginine to produce the monomethyl and dimethylarginine proteins. Bollenbach et al. observed that main sources of methylarginines include proteins responsible for the stability of erythrocyte membrane spectrin-α, spectrin-β and protein 4.1. The same study observed that methylated arginine can change protein function and some peptide chains with methylated arginine induced platelet aggregation [117].

In vitro, upon lysis, erythrocytes are able to release pathologically relevant quantities of free ADMA. After 2 h of incubation free ADMA level increased sevenfold [118]. However, another study suggests that relevant physiological rate of in vivo hemolysis is unlikely to increase significantly human plasma concentration of free ADMA [119]. Further studies need to be undertaken to examine the role of RBC in generation and storage of ADMA, bioavailability and synthesis of NO and its contribution to the pathogenesis of endothelial dysfunction.

DDAH is an enzyme responsible for hydrolysis of both ADMA and L-NMMA, which was found in human RBCs by immunoprecipitation using a specific monoclonal antibody to human DDAH [120]. DDAH activity can be inhibited by SH-specific agents such as inorganic and organic mercury compounds, and by S-nitrosothiols which block the SH group of DDAH that is essential for its hydrolytic activity [121].

## 7. Conclusions

Numerous studies on pathophysiological mechanisms of sleep apnea contributing to an increased cardiovascular risk were conducted. Some clinical studies have shown the importance of molecular changes caused by intermittent hypoxia leading to the endothelial dysfunction. Sleep apnea and all pathophysiological changes could also affect platelets and erythrocytes, which may result in a hypercoagulative state and higher risk of atherothrombotic events.

The enthusiasm for potential clinical usefulness of CPAP therapy is lowered by a deficiency of strong randomized studies confirming its effectiveness in cardiovascular risk reduction. The CPAP improves endothelial function. Nevertheless, when compared with the natural history of OSA, the use of CPAP seems to result in an insignificant reduction in the cardiovascular events incidence, as demonstrated in more than one large clinical trials [111,122]. It is noteworthy that some trend for lowering the risk of stroke for those subjects with good CPAP adherence therapy (≥4 h per night) has been demonstrated, as results from one study [123]. This review proves CPAP effectiveness in different subpopulations of OSA patients and its additional positive effects in reduction of arrythmias, lowering diabetes incidence and better hypertension treatment. 

The discrepancies in the results of clinical studies and some translative difficulties with transposition of the basic research results to clinical setting, may stem from simultaneous role of both, endothelium and blood components in the pathophysiology of the OSA-related thromboembolic complications as well as indicate their common role as the therapeutic target in the course of CPAP treatment.

Therefore, future prospective studies are needed in order to define novel therapeutic strategies aimed at minimizing the cardiovascular risk in subjects with obstructive sleep apnea.

## Figures and Tables

**Figure 1 ijms-22-05139-f001:**
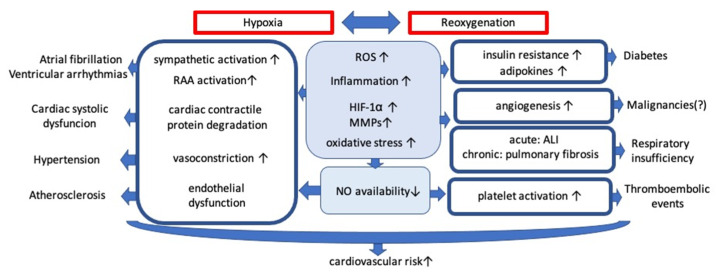
Mechanisms contributing to the progression of cardiovascular disorders. Abbreviations: ROS: reactive oxygen species; HIF-1α: hypoxia inducible factor 1α; MMPs: matrix metalloproteinases; ALI: acute lung injury; RAA: renin-angiotensin-aldosterone; NO: nitric oxide; ↑: increased; ↓: decreased.

**Figure 2 ijms-22-05139-f002:**
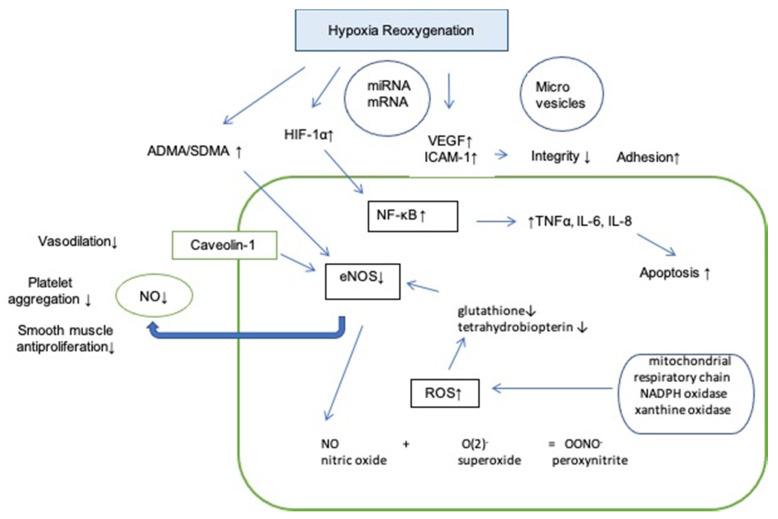
Influence of hypoxia reoxygenation on endothelium. Abbreviations: HIF-1α: hypoxia inducible factor 1 alpha; VEGF: vascular endothelial growth factor; ICAM-1: intercellular adhesion molecule 1; NF- κB: nuclear factor kappa-light-chain-enhancer of activated B cells; TNFα: tumor necrosis factor alpha; IL-6: interleukin 6; IL-8: interleukin 8; SDMA: symmetric dimethylarginine; ADMA: asymmetric dimethylarginine; ROS: reactive oxygen species; eNOS: endothelial nitric oxide synthase; NO: nitric oxide; ↑: increased; ↓: decreased.

**Figure 3 ijms-22-05139-f003:**
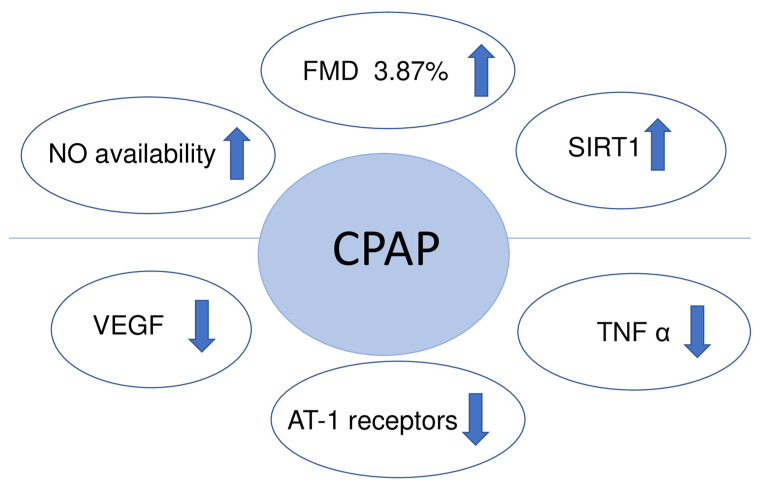
Endothelial function after treatment with CPAP. Abbreviations: VEGF: vascular endothelial growth factor; NO: nitric oxide; FMD: flow mediated dilation; SIRT1: sirtuin 1; TNFα: tumor necrosis factor alpha; AT-1 receptors: expression of angiotensin receptors type-1; ↑: increased; ↓: decreased.

**Figure 4 ijms-22-05139-f004:**
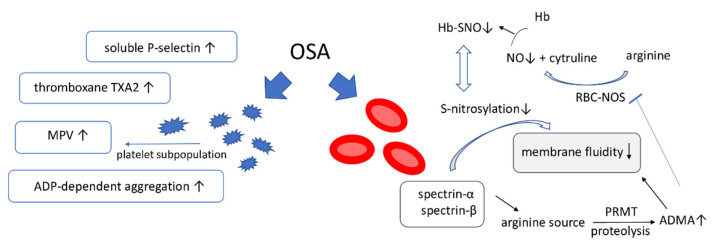
OSA effect on platelets and erythrocytes. OSA: obstructive sleep apnea; MPV-mean platelet volume; ADMA: asymmetric dimethylarginine, PMRT: protein-arginine methyl transferase, Hb-SNO: s-nitrosyloheamoglobin; RBC-NOS: red bleed cell nitric oxide synthase. ↑: increased; ↓: decreased.

## Data Availability

Not applicable.

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
