# Peer review of "Cardiovascular Disorders Triggered by Obstructive Sleep Apnea—A Focus on Endothelium and Blood Components"

_ijms, 2021, doi:10.3390/ijms22105139_

Round 1
Reviewer 1 Report
Dr Mochol and coworkers aims to review the association between obstructive sleep apnea and cardiovascular diseases with focus on endothelium and blood components. The topic is important and there have been several systemic reviews and meta-analyses in this field. In my opinion, this review does not add much to the literature in the current form due to:
1) The paper is too unstructured. It starts with the molecular mechanisms triggered by obstructive sleep apnea and discuss different studies in an unstructured way, and it is not easy to understand the context.
2) The epidemiologic data regarding the association between obstructive sleep apnea and different cardiovascular diseases as well as confounding factors and comorbidities are mentioned but without any information about molecular mechanisms or biomarkers in those subgroups.
3) If this paper aims to be a systematic review, the authors should register the study at the The International Prospective Register of Systematic Reviews (PROSPERO) and follow the guidelines accordingly.
4) There are no tables given to summary the key studies to illustrate the publication year, the number of patients included, the study type (cross-sectional/prospective/cohort study/RCT) and the main findings.
Reviewer 2 Report
I have read with interest the review by Mochol and co-authors which describes the role of OSA related endothelium and blood component alterations in the development of CV diseases. The review is interesting but not so easy to read particularly for clinicians who are less familiar with biomolecular mechanisms. Most importantly the structure of the review should be more focussed on the main topic: the authors start with discussing the mechanisms of endothelial dysfunction but then they presented some data on OSA in several subgroups of patients without keeping the same topic and losing the focus (eg. 4.6. Cardiovascular risk). I would discuss in the present review about endothelium and blood component alterations only as some other mechanisms are more consolidated.
As this seems a narrative review I suggest to adopt a clear framework (eg. https://www.elsevier.com/__data/promis_misc/ANDJ%20Narrative%20Review%20Checklist.pdf)
I have some additional comments for the authors which might further improve the work:
- please consider shortening the text and add instead figures and/or tables, this might improve readability - please double check reference #30: authors are not properly displayed- Abstract (and introduction): "Obstructive sleep apnea (OSA) is known to be an independent cardiovascular risk factor, but the exact molecular mechanisms underlying the progression of cardiovascular disorders remain 12poorly understood" I would rephrase this. After more than 30 years of research on this topic we have understood most of the pathophysiological mechanisms like intermittent hypoxia, arousal from sleep and enhanced sympathetic activation. Nevertheless, less is known about blood components which justifies the current review.- Introduction: "Obstructive sleep apnea (OSA) is characterized by recurrent obstruction of the upper airways" please amend to "airway" Same line: Sleep deprivation occurs when a person is not able to get enough sleep thus please delete "deep sleep deprivation"
- please describe clearly all acronyms throughout the text when you mention them for the first time- the chapter "Molecular consequences of Hypoxia/Reoxygenation (H/R) on endothelial function in OSA before CPAP treatment" should be more focussing in connecting the dots between the more consolidated mechanisms (sympathetic activation) and the less studied ones (ROS overproduction and cellular damage). Again, a figure or table could help. - introduction " An understanding of the 53biology of NO, O2- and ONOO- is necessary to resolve this controversy" not clear what is the controversy, please clarify- this reviewer believes that a figure showing step by step the alterations involving endothelium and blood component that connect OSA to CV diseases would be helpful- the manuscript deserves an English language revision and a typo check (eg. assess rather than asses (page 4, line 160)- consider adding a table showing all studies on OSA patients focussing on endothelial function alterations induced by CPAP - the title "Comorbidities and treatment effectiveness" is a bit vague. Consider something like: Endothelial function after treatment with CPAP in specific subgroups- page 7: "Although it is well known that the treatment of sleep apnea lowers blood pressure, the effectiveness of the therapy remains low" This is debatable, the last systematic review and meta-analysis (doi: 10.1183/13993003.01945-2019) showed a net overall BP reduction of 2 mmhg which, translated into CV events and mortality reduction over the long term, is not completely negligible.- references are not always appropriately cited: eg. page 7: "CPAP was associated with 323significant reductions in 24-hour ambulatory systolic blood pressure of 2.32 mm Hg and 324diastolic blood pressure of 1.98 mm Hg[70-71]" Ref #70 does not seem appropriate. Please double check all references in the text and confirm their appropriateness according to the context- The systematic review cited above (doi: 10.1183/13993003.01945-2019) identified some predictors of BP reduction under CPAP: one of this is severe hypoxia but not AHI. This could reinforce the concept that reducing ROS production could lead to a better cardiovascular improvement? Please discuss- conclusions are too short and not well balanced considering the length of the whole manuscript.
Round 2
Reviewer 1 Report
The paper is now improved, and more structured. The narrative character of the review is clarified.
Given the ongoing debate regarding the association between OSA and myocardial infarction, and the neutral results of CPAP treatment in the recent randomized controlled trials (RICCADSA, SAVE, ISAACC), it could be valuable to add a section "Myocardial Infarction", and review the literature in the context of endothelial biomarkers and molecular mechanisms involved in ischemic preconditioning. These studies can be added in the Tables, respectively.
Author Response
Reviewer #1:
We would like to thank the Reviewer for an in-depth analysis of the manuscript and for pivotal comments provided which have resulted in a significant improvement of this manuscript.
- Given the ongoing debate regarding the association between OSA and myocardial infarction, and the neutral results of CPAP treatment in the recent randomized controlled trials (RICCADSA, SAVE, ISAACC), it could be valuable to add a section "Myocardial Infarction", and review the literature in the context of endothelial biomarkers and molecular mechanisms involved in ischemic preconditioning. These studies can be added in the Tables, respectively.
We agree with the Reviewer that adding the comment on the assosciation between OSA and MI is a matter of an interesting debate and the comments on that issue were missing in the. Hence, according to the Reviewers’ suggestion we have added the following:
4.2 Myocardial infarction
Hypoxia-reoxygenation (H/R) injury is observed during an early phase of myocardial infarction and at the beginning of reperfusion therapy as well as in the severe OSA desaturations, which mimics asphyxia (hypoxia). The increased peroxynitrite (ONOO-) formation in the heart during H/R, may change several proteins which compose the contractile machinery of the heart leading to systolic dysfunction and cardiac injury [53]. Peroxynitrites formed during the injury may lead to the nitration of cardiac contractile proteins (including the myosin light chain 1 and 2, MLC1 and MLC2) leading to their increased susceptibility to subsequent proteolysis by the matrix metalloproteinases (i.e. MMP-2) [121, 122]. Similar changes were observed in an ex vivo model of myocardial infarction [123], pointing thus at the same cardiotoxic pathophysiological mechanisms of the ischemia/reperfusion and hypoxia/reoxygenation cardiac injury. In that studies, the compensatory increase in tissue inhibitor of matrix metalloproteinases-4 (TIMP-4) expression was observed during ischemia, but not reperfusion, which reflects its role in the ischemic precontitioning.
In clinical setting, the prevalence of moderate/severe OSA in patients with coronary artery disease was assosiateted with diminished plasma level of C1q/TNF-related protein-9 (CTRP9) [54]. CTRP9 is an adipokine that protects the heart against ischemic injury and ameliorates cardiac remodeling, which in turn could explain the role of OSA in exacerbating the coronary artery disease. The role of OSA in increased incidence of myocardial infarction can be also considered together with the progression of atherosclerosis and altered platelet function. The role of CPAP effectiveness in reducing the myocardial infarction incidence is not proven yet. The ISAACC study showed that among non-sleepy patients with acute coronary syndrome, the presence of OSA was not associated with an increased prevalence of cardiovascular events and treatment with CPAP did not significantly reduce this prevalence.[55] On the contrary, RICCADSA trial confirmed that CPAP treatment may reduce this risk, if the device is used at least 4 h/day.[56]
Table 2. Cardiovascular thromboembolic disorders and CPAP effectiveness.
|
DEMONSTRATED PLATELET-DERIVED MOLECULAR PATHOMECHANISM |
THROMBOEMBOLIC DISORDERS |
CLINICAL EFFECT OF OSA TREATMENT-BASED ON CLINICAL STUDIES |
|
-platelet hyperaggregability for both ADP and collagen
-lower inhibitory rate of the ADP-dependent aggregation
- high residual platelet reactivity after acetylsalicylic acid or clopidogrel therapy [95] |
Stroke |
Reduction in risk of stroke in elderly patients[107] (2021, retrospective cohort study, 5757 patients) |
|
Myocardial Infarction |
No significant effect in the MI incidence reduction in prospective observation of the CPAP treatment patients [55]( 2020, randomized controlled trial, 2551 patients), [108] (2016, randomized controlled trial, 2717 patients) |
Table 1 Endothelial function after treatment in specific subgroups.
|
OSA SUBPOPULATION |
DEMONSTRATED MOLECULAR PATHOMECHANISM |
EFFECT OF OSA TREATMENT |
|
Myocardial Infarction |
Increased peroxynitrite formation [53], nitration of cardiac contractile proteins (MLC1 and MLC2) and their subsequent degradation (by MMP-2) [121, 122], protective role of TIMP-4 in ischemic preconditioning [123] |
The ISAACC - among non-sleepy patients with acute coronary syndrome, treatment with CPAP did not significantly reduce thieprevalence of acute coronary syndromes.[55] (2020, randomized controlled trial, 2834 subjects) On the contrary, RICCADSA trial confirmed that CPAP treatment may reduce this risk, if the device is used at least 4 h/day.[56] (2016, randomized controlled trial, 245 subjects) |
Reviewer 2 Report
Figures have certainly improved readibility of the manuscript. However, the authors adressed only partially my previous comments therefore I strongly encourage them to sumbit a point by point reply.
Also it appears references were not updated as per my comment
- references are not always appropriately cited: eg. page 7: "CPAP was associated with 323 significant reductions in 24-hour ambulatory systolic blood pressure of 2.32 mm Hg and 324 diastolic blood pressure of 1.98 mm Hg[70-71]" Ref #70 does not seem appropriate. Please double check all references in the text and confirm their appropriateness according to the context
Other points were not discussed such as
- The systematic review cited above (doi: 10.1183/13993003.01945-2019) identified some predictors of BP reduction under CPAP: one of this is severe hypoxia but not AHI. This could reinforce the concept that reducing ROS production could lead to a better cardiovascular improvement? Please discuss
Author Response
Following your letter concerning our submission number ijms-1195260
entitled “Cardiovascular disorders triggered by obstructive sleep apnea –
focus on endothelium and blood components” we proceeded to a deep analysis of our manuscript.
The Editor’s and Reviewers’ comments have been taken into high account and revealed to be an important base of support for restructuring our manuscript. We performed a reorganization of the manuscript and, in our opinion, significant improvement of the manuscript has been made which supports clearly our results. In response to the comments and concerns we now include the following:
Reviewer #2:
We would like to thank the Reviewer for an in-depth analysis of the manuscript and for pivotal comments provided which have resulted in a significant improvement of this manuscript.
- The review is interesting but not so easy to read particularly for clinicians who are less familiar with biomolecular mechanisms.
According to the suggestions, we have summarized the data in the Tables with respect to particular cardiovascular disorder and secondly with respect to potential platelet-derived molecular patho-mechanism contributing to OSA-related thromboembolic event, which was done in the first resubmission of the manustript. Moreover, according to the Reviewer’s suggestion, now we discuss more precisely in our review about endothelium and the peroxynitrite derived oxidative stress:
4.2 Myocardial infarction
Hypoxia-reoxygenation (H/R) injury is observed during an early phase of myocardial infarction and at the beginning of reperfusion therapy as well as in the severe OSA desaturations, which mimics asphyxia (hypoxia). The increased peroxynitrite (ONOO-) formation in the heart during H/R, may change several proteins which compose the contractile machinery of the heart leading to systolic dysfunction and cardiac injury [53]. Peroxynitrites formed during the injury may lead to the nitration of cardiac contractile proteins (including the myosin light chain 1 and 2, MLC1 and MLC2) leading to their increased susceptibility to subsequent proteolysis by the matrix metalloproteinases (i.e. MMP-2) [121, 122]. Similar changes were observed in an ex vivo model of myocardial infarction [123], pointing thus at the same cardiotoxic pathophysiological mechanisms of the ischemia/reperfusion and hypoxia/reoxygenation cardiac injury. In that studies, the compensatory increase in tissue inhibitor of matrix metalloproteinases-4 (TIMP-4) expression was observed during ischemia, but not reperfusion, which reflects its role in the ischemic precontitioning.
In clinical setting, the prevalence of moderate/severe OSA in patients with coronary artery disease was assosiateted with diminished plasma level of C1q/TNF-related protein-9 (CTRP9) [54]. CTRP9 is an adipokine that protects the heart against ischemic injury and ameliorates cardiac remodeling, which in turn could explain the role of OSA in exacerbating the coronary artery disease. The role of OSA in increased incidence of myocardial infarction can be also considered together with the progression of atherosclerosis and altered platelet function. The role of CPAP effectiveness in reducing the myocardial infarction incidence is not proven yet. The ISAACC study showed that among non-sleepy patients with acute coronary syndrome, the presence of OSA was not associated with an increased prevalence of cardiovascular events and treatment with CPAP did not significantly reduce this prevalence.[55] On the contrary, RICCADSA trial confirmed that CPAP treatment may reduce this risk, if the device is used at least 4 h/day.[56]
- We have double checked the references, and corrected the ref. #30, as follows:
- Thunström E, Glantz H, Yucel-Lindberg T, Lindberg K, Saygin M, Peker Y. CPAP Does Not Reduce Inflammatory Biomarkers in Patients With Coronary Artery Disease and Nonsleepy Obstructive Sleep Apnea: A Randomized Controlled Trial. Sleep. 2017 Nov 1;40(11). doi: 10.1093/sleep/zsx157. Erratum in: Sleep. 2019 Feb 1;42(2): PMID: 29029237.
- We woluld like to thank the Reviewer for suggestion to comment on the meta-analysis: Pengo MF, Soranna D, Giontella A, Perger E, Mattaliano P, Schwarz EI, Lombardi C, Bilo G, Zambon A, Steier J, Parati G, Minuz P, Fava C. Obstructive sleep apnoea treatment and blood pressure: which phenotypes predict a response? A systematic review and meta-analysis. Eur Respir J. 2020 May 7;55(5):1901945.
This interesting review analyzes potential predictors of BP response on the CPAP treatment. We have commented on this issue, as follows:
“.. intermittent hypoxia, and not recurrent arousals, appears to be the dominant cause of daytime increases in blood pressure in OSA. There is a need to better understand the effects of CPAP on BP since a better characteristics of patients, who benefit most might help to tailor therapy according to the expected benefit to reduce BP and in general improve the patient‟s cardiovascular risk profile. Interestngly, a systematic review and meta-analysis of randomized controlled trials (RCTs) compared the effect of CPAP on BP in particular subgroups of patients and aimed at defining the group with the best response to treatment. The meta-analysis has defined younger age, uncon-rolled blood pressure and severe OSA-related oxygen desaturations as positive predictors of a favourable blood pressure response to OSA treatment. Desaturation of less than 77% were associated with a greater BP drop at follow-up in treated patients further supporting the role of intermittent hypoxia in the pathogenesis of OSA- related hypertension [123]. This oobservation points at oxidative stress as on the important therapeutic target”
- We have read the manuscript thoroughly, and all the abbreviations we have noticed, are now expanded.
- We believe that the Reviewer made a valid point when stating that the sentence: “An understanding of the biology of NO, O2- and ONOO- is necessary to resolve this controversy” seems to be incomplete. Indeed, it was an excessive shortcut, we apologize for this error, which we have now fixed as follows:
“An understanding of the biology of NO, O2- and ONOO- as well as the importance of the nitric oxide synthase uncoupling in physiological and pathological setting are crucial for understanding the nitrosative stress-related controversies. There is a critical balance between cellular concentrations of NO, O2-, and superoxide dismutase, which physiologically favours NO production but in pathological conditions such as ischemia/reperfusion(I/R) results in ONOO- generation.”
- We have taken into account the Reviewer’s comment indicating that the conclusions were too short and not well balanced considering the length of the whole manuscript, which resulted in an improvement of this section.
It is our strong belief that the International Journal of Molecular Sciences is the ideal vehicle to communicate our translational analysis of the molecular consequences of OSA and its treatment with CPAP on endothelium, platelets and red blood cells, presented in this manuscript. Therefore, we decided to resubmit our manuscript to your journal, alongside this letter, with the conviction that there is significant improvement of the manuscript, both scientifically and structurally.
Sincerely yours,
Adrian Doroszko, MD PhD
Round 3
Reviewer 1 Report
The paper is now further improved.
There are some minor typos that can be corrected in the final publication phase.
In Table 1, the number of participants included in the RCT arms were 1264 in the ISAACC trial (not 2834), and 244 in the RICCADSA trial (not 245).
Author Response
Wroclaw, May 7th, 2021
Assistant Editor
“International Journal of Molecular Sciences”
Dear Prof. Codruta Cormos,
Following your letter concerning our submission number ijms-1195260
entitled “Cardiovascular disorders triggered by obstructive sleep apnea –
focus on endothelium and blood components”, we proceeded to analysis of our manuscript. In response to the comments and concerns we now include the following:
Reviever #1:
- We would like to thank the Reviewer for pointing out the errors in the Table 1 regarding the numbers of participants included in the ISAACC and the RICCADSA trials. We have corrected the numbers, as appropriate.
2. We have read the manuscript thoroughly, and all the typos we have noticed are now corrected
It is our strong belief that the International Journal of Molecular Sciences is the ideal vehicle to communicate our translational analysis of the molecular consequences of OSA and its treatment with CPAP on endothelium, platelets and red blood cells, presented in this manuscript.
Sincerely yours,
Adrian Doroszko, MD PhD